# Quantifying the spatiotemporal dynamics of the first two epidemic waves of SARS-CoV-2 infections in the United States

Rafael Lopes[1,2]*, Yu Lan[1,2], Melanie H. Chitwood[1,2], Fayette Klaassen[3], Joshua A. Salomon[4], Nicolas A. Menzies[3], Joshua L. Warren[2,5], Nathan D. Grubaugh[1,2], Ted Cohen[1,2], Nicole A. Swartwood[3‡]*

**1** Department of Epidemiology of Microbial Diseases, Yale School of Public Health, Yale University, New Haven, Connecticut, United States of America, **2** Public Health Modeling Unit, Yale School of Public Health, Yale University, New Haven, Connecticut, United States of America, **3** Department of Global Health and Population, Harvard T.H. Chan School of Public Health, Harvard University, Boston, Massachusetts, United States of America, **4** Department of Health Policy, Stanford University, Stanford, California, United States of America, **5** Department of Biostatistics, Yale School of Public Health, Yale University, New Haven, Connecticut, United States of America

‡ This author are senior author on this work.
* rafael.lopes@yale.edu (RL); nswartwood@hsph.harvard.edu (NAS)

## Abstract

SARS-CoV-2 infection rates displayed strikingly organized patterns of temporal and spatial spread as new variants were introduced and subsequently transmitted within the United States. While these spatio-temporal "waves" of infection have been described previously, attempts to quantify the speed and extent of these waves have been limited. Here, we estimate and compare the wavefront speed and spatial expansion of the first two major infection waves in the United States, illustrating these dynamics through detailed visualizations. Our findings reveal that the origins of these waves coincide with large gatherings and the relaxation of masking mandates. Notably, we found that the second wave spread more rapidly than the first, possibly driven by multiple introduction events. These analyses highlight regional heterogeneity in epidemic dynamics and underscore the importance of localized public health measures in mitigating ongoing outbreaks.

## Author summary

In this work, we developed a set of tools and methods designed to produce high-resolution visualizations, as well as to quantify the complex and intricate patterns associated with the spatial and temporal spread of SARS-CoV-2 in the United States of America. Our focus was on examining and characterizing the dynamics of transmission as they unfolded during the first two major waves of infections that were observed across the contiguous United States territory. In developing such tools, we were able to generate detailed visual representations

which permits unrestricted use, distribution, and reproduction in any medium, provided the original author and source are credited.

**Data availability statement:** Daily county estimates of per 100,000 infections used in the analysis are available at Harvard Dataverse https://doi.org/10.7910/DVN/G2ZXJG. The relevant scripts and necessary origination data, such as hexagonally distributed infections per 100,000 are publicly available at https://github.com/covidestim/waves. These materials allow the reproduction of all the figures and analytical products in this analysis. The static repository of all codes and intermediary data products are available on Zenodo: https://zenodo.org/records/18624632.

**Funding:** This project is supported by Cooperative Agreement NU38OT000297 from the Centers for Disease Control and Prevention (CDC) the Council of State and Territorial Epidemiologists (CSTE) (NAS, FK, JAS, TC and NAM), SHEPheRD Contract 200-2016-91779 from the CDC, and the CDC Broad Agency Announcement Contract 75D30122C14697 (RL, YL, MHC, NDG, JLW, and TC). This work does not necessarily represent the views of the CDC or CSTE. The funders had no role in study design, data collection and analysis, decision to publish, or preparation of the manuscript.

**Competing interests:** I have read the journal's policy and the authors of this manuscript have the following competing interests: NDG is a paid consultant for BioNTech. All the other authors have declared that no competing interests exist.

of how the virus had moved and spread geographically over time, while also providing a means to numerically capture and describe the speed of those spreading patterns. The approaches we developed were applied comprehensively across the lower 48 states of the USA, allowing us to track and document how SARS-CoV-2 propagated across different regions during each of the two significant waves of infection that defined the early period of the pandemic in those regions.

## Introduction

Person-to-person transmission of SARS-CoV-2 was confirmed in the United States (US) in late January 2020, and a national public health emergency was declared six weeks later [1,2]. A distinct feature of the subsequent SARS-CoV-2 epidemic, viewed from a national level, was the appearance of several periods of rapid increase and slower decline in estimated infection rates. These epidemic "waves" largely coincided with the introduction of genetically distinct SARS-CoV-2 variants [3,4] and were also affected by changes in the use and effectiveness of infection-mitigating interventions [5,6] and changes in population movement [7–10].

Here we quantify and compare the speed of wavefront expansion and the areal extent of wave spread over distinct epidemic waves. We apply a version of the Besag, York, and Mollié (BYM) spatial model [11,12] to produce temporally- and spatially-resolved estimates of SARS-CoV-2 infection rates [13] over the first two major waves of infections and use these estimates to estimate the speed of movement and spatial extent of wave involvement for the contiguous United States.

The two major epidemic waves we analyzed overlap with the introduction of the 'Alpha' (B.1.1.7) variant of concern (VOC) in mid-September 2020 and the 'Delta' (B.1.617.2) VOC in early July 2021 [14–18]. We note, however, that the first wave we study begins to rise in advance of the widespread emergence of the Alpha VOC, hence we refer to these waves as Wave 1 and Wave 2, as they do not perfectly align with a singular genetic variant in circulation [17–20].

We note that, in addition to being associated with distinct viral variants, the two waves considered in this study were spreading in conditions that also differed in terms of levels of population immunity (conferred by either previous infections or vaccinations) and the types of non-pharmaceutical interventions being employed. Wave 1 occurred before vaccines were available, and when immunity was acquired as a result of infections with the wild-type variants [21,22]. Wave 1 also began during a time of nationwide travel restrictions and widespread policies mandating mask use in congregate settings [5,7,9].

In contrast, wave 2 occurred when there were fewer mobility restrictions and fewer mandatory masking policies in place, though these policies varied markedly by geography within the US. Vaccines had also become available in January 2021 in the US, producing higher levels of apparent population immunity for wave 2 compared

to wave 1 [21,22]. We note wave 2 was largely attributable to the 'Delta' variant, which was characterized by its ability to escape immune responses acquired through either previous infection or vaccination.

## Materials and methods

### Data sources

We used previously published daily estimates of SARS-CoV-2 infections at the county level in the United States from a Bayesian nowcasting model that synthesized reported COVID-19 cases, hospitalizations, and deaths, accounting for both under ascertainment and time lags in disease progression [13]. Our dataset encompasses the period from March 2020 to December 2021. To estimate population denominators for our per 100,000 analyses, we used 2019 United States population estimates via Meta, which provided population size estimates at a resolution of 30 square meters [23].

### Regularization of geographical units

Counties were the smallest geographical unit with consistently available data for COVID-19 cases, hospitalizations, and deaths in the United States, which dictated the geographic resolution at which SARS-CoV-2 infections could be estimated. However, as the geographic size and population density of counties differ systematically across the US (larger area and sparser population density in the West than in the East), a county-level analysis would bias estimates of the wavefront speed and area of spread.

To improve our ability to detect spatial patterns in the data that were robust to administrative differences in county size, we first created a spatially regularized grid to which we allocated population and infections per 100,000 across the contiguous US. The grid comprises 7,517 unique hexagons, each enclosing 1,100 square kilometers. We employed an area-weighted approach to distribute population from the 30 square-meter estimates to the hexagons (**Fig 1A-1C**), assuming a constant population throughout the analysis period. Our daily estimates of SARS-CoV-2 infections per 100,000 for each hexagon were produced by the distribution of estimated infections from counties to hexagons [22] based on the fraction of each county's population contained in each hexagon, assuming that per 100,000 infection rates were distributed equally within a county. (**Fig 1D-1E**).

### Assessment of spatial patterns of infections per 100,000

We fit a modified version of the BYM model as implemented in R-INLA [24], called BYM2, to estimate spatially smoothed rates of SARS-CoV-2 infections per 100,000 of population across the hexagonal grid. The BYM2 is a reparametrized version of the original Besag [12] model, which improves the assignment of prior distributions for the corresponding model parameters and the subsequent interpretation of parameters [11]. Specifically, our model is given as

$$Z(A_i) = \mu + \theta(A_i) + \epsilon(A_i),$$ (1)

where $Z(A_i)$ the infections per 100,000 in hexagon $A_i$, $\mu$ is the global intercept, $\theta(A_i)$ are random effects that are assigned the BYM2 prior distribution, and $\epsilon(A_i)$ are residual error terms assumed to be statistically independent. For more details on the model and priors used, see the S1 Appendix. We fit the equation 1 model separately for each day over March 2020 to December 2021. We produced a sequence of these daily spatial distributions, which we then used to calculate the speed of wave front movement and spatial expansion and to visualize changes over time. We opted for a separate model fit on each day to avoid temporal over smoothing effects, as the infection estimates were derived from a nowcasting model that already applied temporal smoothing.

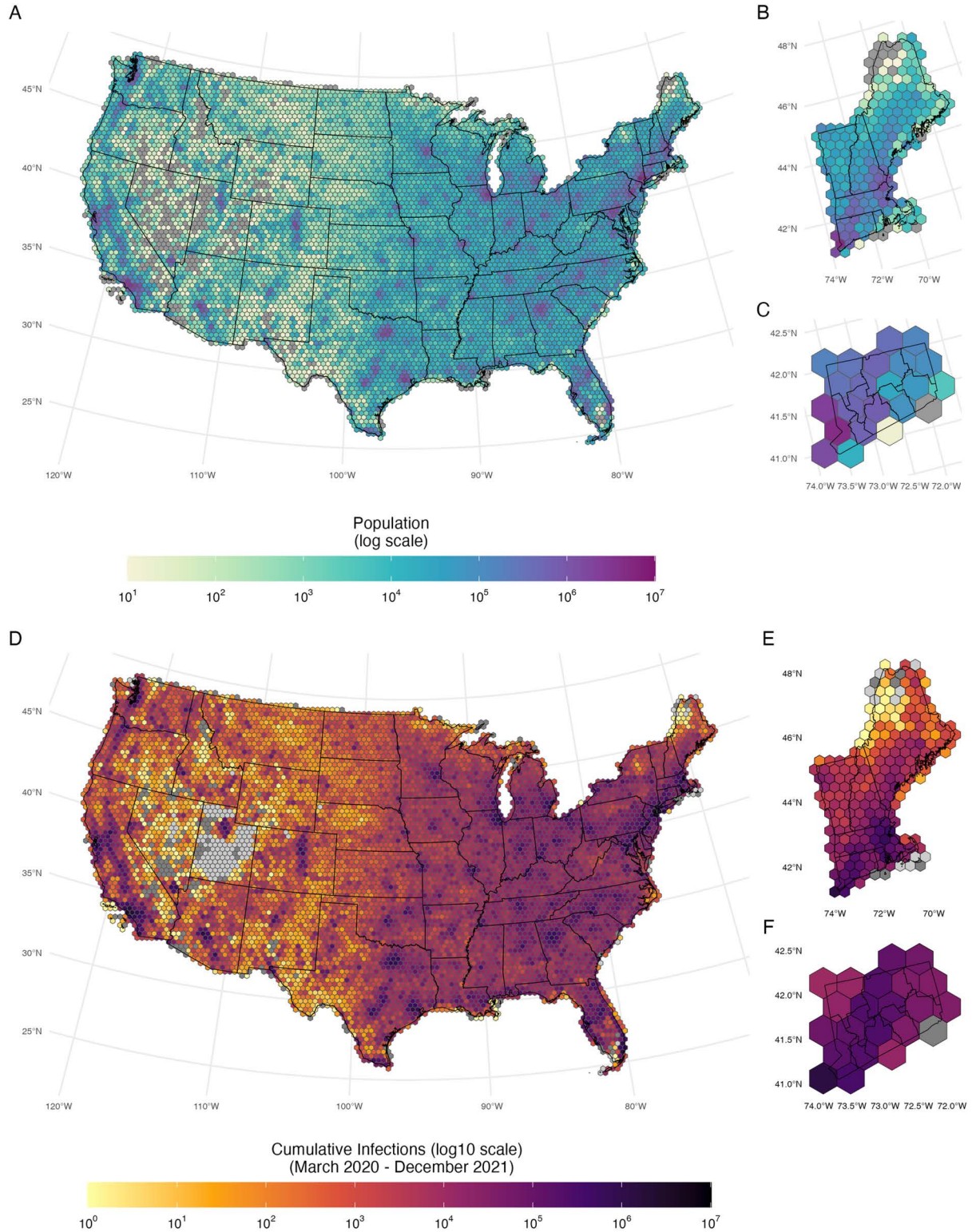

**Fig 1. United States population and estimated cumulative SARS-CoV-2 infections per 100,000 distributed across the hexagonal grid.** Panel **A, B,** and **C**: United States', New England's, and Connecticut's Meta 30m population estimates on the hexagonal grid. Panel **D, E,** and **F**: United States, New England, and Connecticut cumulative infections per 100,000 persons on the hexagonal grid (March 2020–December 2021). Grey hexagons

indicate no population (panels **A** to **C)** and no infections (panels **D** to **F**), void hexagons indicate no infections ever estimated (panels **D** to **F**). All maps were generated using United States, state, and county borderlines maps in the public domain from the Census Bureau, which were downloaded through the R package Tigris [30]. The shapefile generated for this analysis with the population estimates and cumulative infection estimates can be found at: https://github.com/covidestim/waves/tree/waves-manuscript/Data/data-products. Note: Numbers are given in a log 10 scale.

### Definition of a wave

To formally determine hexagons within a wave, we defined a threshold of 190 SARS-CoV-2 infections per 100,000 persons or more, which we coded as being part of an infection wave. Any hexagon within a wave could leave the wave if infections fell below that threshold. For our main analysis, we used the above-mentioned threshold of 190 daily infections per 100,000 persons to characterize a hexagon as being within a wave. This value represents the 75th percentile of the infections-per-100,000 value distribution during the study period (S1 Fig). We conducted sensitivity analyses using 50th (127 infections per 100,000) and 90th (233 infections per 100,000) percentiles of infections per 100,000 persons as alternative threshold values (S2 Fig). To allow for multiple infection centers, we did not limit our definition to a single contiguous set of hexagons.

Using this definition, we categorized Wave 1 as the sets of contiguous hexagons above the threshold within the period ranging from September 17, 2020 to February 19, 2021, and Wave 2 as the sets of contiguous hexagons above the 190 infections per 100,000 threshold during the period from July 3, 2021 to September 4, 2021. We identified the epicenters of each wave by visual inspection.

### Calculation of wavefront speed and areal wave expansion

We first identified the leading edge, or wavefront, of each wave. This wavefront was defined as the two-dimensional boundary between hexagons included and excluded from a wave on each day of the analysis. After delineating the wavefront, we estimated two quantities: the daily areal wave expansion and the daily wavefront speed. To calculate the areal wave expansion rate, we tracked the daily change in the number of hexagons included in a wave and multiplied this by the area of each hexagon (1,100 km$^2$) to calculate the expansion rate in km$^2$. This quantity is the difference between the area of the wave at day $d+1$ and day $d$, giving an areal wave expansion rate measured in km$^2$/day.

To estimate the wavefront speed, we identified the vertices of the line-segments comprising the wavefront on each day, $d$. We calculated the distance between each vertex and the nearest point on the subsequent wavefront at day $d+1$, which provides a measure of distance traveled per day (km/day), or speed for each vertex of the wavefront. This allows us to capture the heterogeneity in speed along the length of the wavefront. For each day, we estimated the daily mean speed as the mean of all wave vertex speeds estimated for that day. We also summarized the speed of each wave by calculating the mean of the estimated speed for all days and wave vertices with equal weights. Finally, we estimated the length of the wavefront that was moving at a rate greater than the median of the daily maximum estimated speeds. This length was calculated as the sum of the line-segment lengths adjacent to a vertex with an estimated speed greater than the median daily maximum speed.

### Statistical analysis

All analyses were performed using R Statistical Software (v4.3.0) [25]. We used the **INLA**, **sf**, **sp**, **spdep**, **areal**, and **magick** R packages [24,26–30]. All the maps were generated using United States, state, and county borderlines maps in the public domain from the Census Bureau, which were downloaded through the R package Tigris [31,32].

### Results

#### Spatiotemporal patterns of SARS-CoV-2 infections across the United States

As described in the **Materials and Methods**, we categorized infections as belonging to Wave 1 (September 17, 2020 to February 19, 2021) or Wave 2 (July 3, 2021 to September 4, 2021) (**Fig 2A**). **Fig 2B**-**2I** shows the smoothed estimates of SARS-CoV-2 infections per 100,000 from the BYM2 model on eight dates, leading up to the peak of each of the two

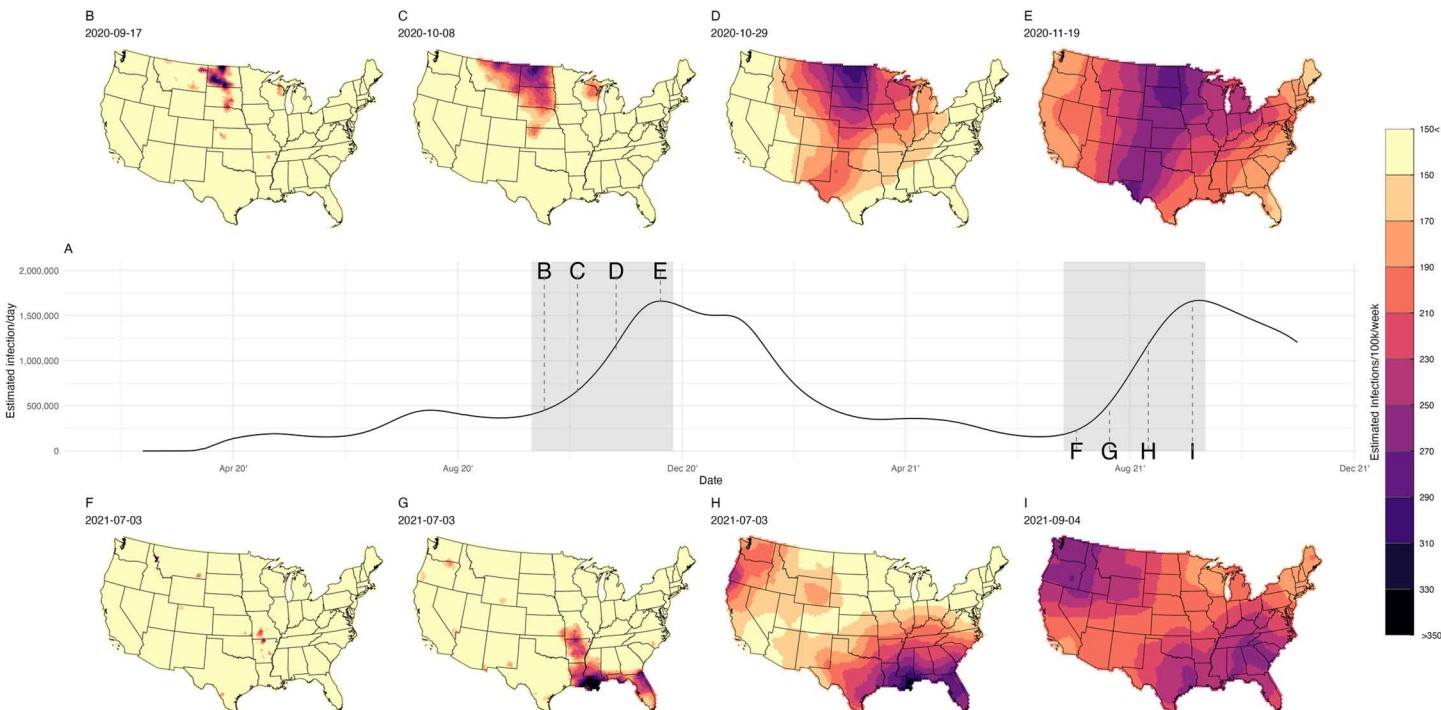

**Fig 2. Estimated infections per 100,000 of SARS-CoV-2 in the United States, March 2020–December 2021.** Panel **A**: Time series of SARS-CoV-2 infection estimates for the United States; the gray shaded areas show the first two large waves of infections. Panels **B, C, D,** and **E**: Sequence of the spatially smoothed estimates of SARS-CoV-2 infections per 100,000 associated with Wave 1 at 4 time points. Panels **F, G, H,** and **I**: Sequence of the spatially smoothed estimates of SARS-CoV-2 infections per 100,000 associated with Wave 2 at 4 time points. All maps were generated using United States, state, and county borderlines maps in the public domain from the Census Bureau, which were downloaded through the R package Tigris [30]. The shapefile generated for this analysis with the population estimates and cumulative infection estimates can be found at: https://github.com/covidestim/waves/tree/waves-manuscript/Data/data-products. For a more detailed visualization of each wave, see the S1-S2 Movies.

waves analyzed. Wave 1 originated with a set of hexes spanning central South Dakota, eastern North Dakota, and north-eastern Montana (**Fig 2B**) in mid-September of 2020. Then the wave extended south and to both coasts, achieving a peak of over 1.5 million prevalent infections/day by mid-November 2020 (**Fig 2A** and **2C**-**2E**). Wave 2 originated in the Ozarks (southern Missouri and northern Arkansas) (**Fig 2F**) in early July 2021 and then expanded further south (**Fig 2G**-**2I**). Secondary epicenters of infection appeared later in July in the Pacific Northwest (**Fig 2G**), and the infection wave subsequently spread throughout the western United States (**Fig 2G**-**2I**). This wave had a peak of 300 per 100,000 infections/day by early September 2021.

Animations of the infection waves are provided in S1 and S2 Movie **files,** which present maps of the daily, spatially smoothed estimated infections for waves 1 and 2, respectively. All maps were generated using United States, state, and county borderlines maps in the public domain from the Census Bureau, which were downloaded through the R package Tigris [30]. The shapefile generated for this analysis with the population estimates and cumulative infection estimates can be found at: https://github.com/covidestim/waves/tree/waves-manuscript/Data/data-products.

## Wave expansion

Fig 2B-2C shows contour plots of the speed of expansion for each wave. Wave 1 expanded from an area of 134,200 km² to 6,515,300 km² between September 8, 2020 and November 11, 2020 (Fig 3, **'1st Wave' bars**). Wave 2 expanded from

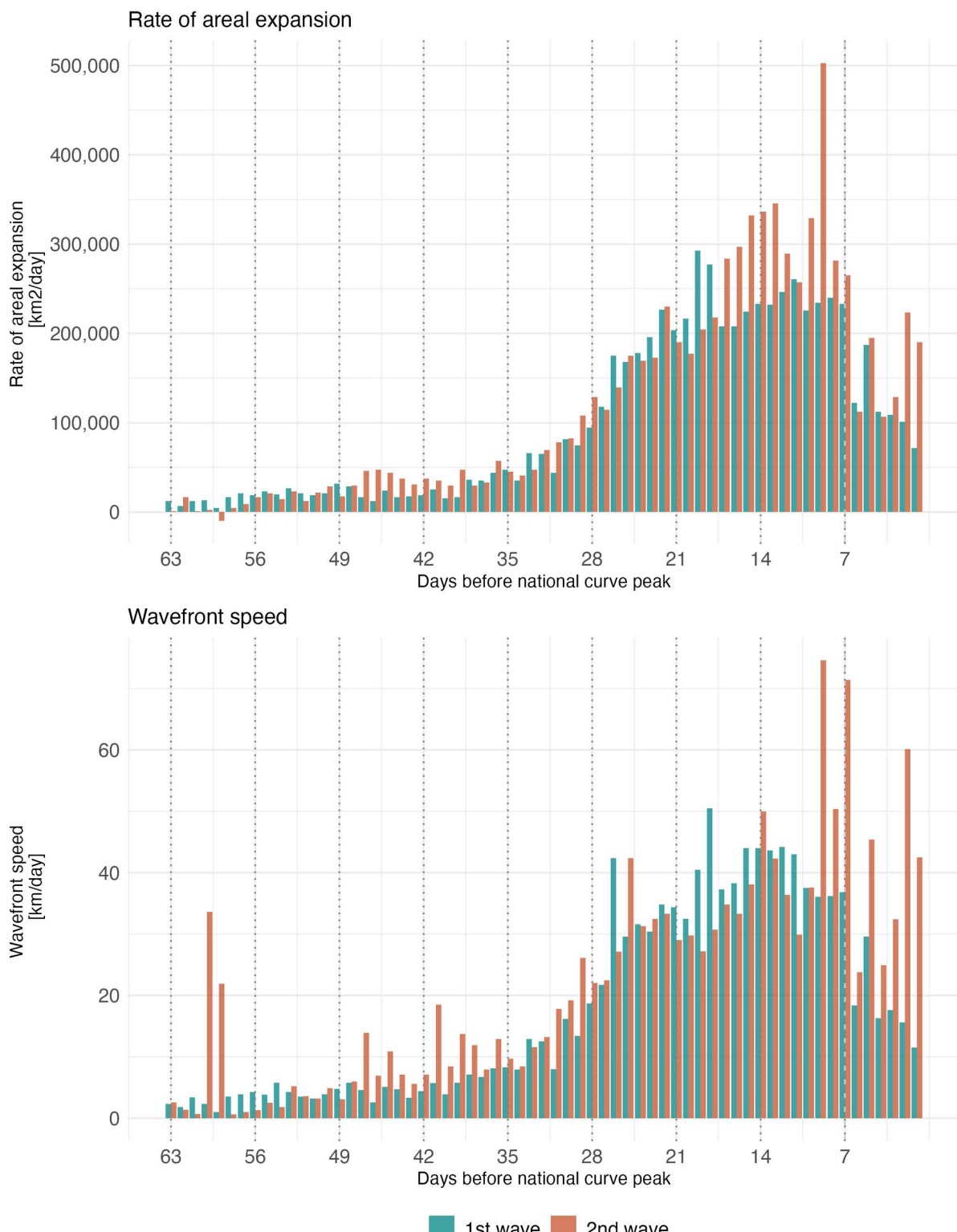

**Fig 3. Daily mean areal expansion rate and wavefront speed for SARS-CoV-2 infection waves for 63 days preceding the infection per capita peak.** (Wave 1: September 8, 2020–November 11, 2020; Wave 2: July 7, 2021–September 4, 2021). **Top Panel:** Mean daily areal expansion rate. **Bottom Panel:** Mean daily wavefront speed. Note: Areal expansion rate is the daily change in wave area. Wavefront speed is the distance from a vertex of the hexagonal grid at day d to the nearest point on the line at day d + 1.

an area of 23,100 km$^2$ to 7,573,500 km$^2$ between July 7, 2021 and September 4, 2021 (Fig 3, '2$^{nd}$ Wave' bars). The wave-like patterns were robust to the choice of threshold (S2 and S3 Figs).

Fig 3 compares the areal expansion of each wave for each analyzed day before each wave's respective peak. Wave 1 had a mean areal expansion rate of 101,287 km$^2$/day (median= 64,900 interquartile range (IQR): 20,350–199,650 km$^2$/day). Wave 2 had a slightly higher mean areal expansion rate of 119,848 km$^2$/day (median = 69,300; IQR = 29,700–192,500 km$^2$/day) during the analysis period.

### Wave speed

The overall mean wavefront speed of Wave 1 was 20.3 km/day (median = 20.6; IQR = 0–35.6 km/day) across all geographies and the 63 days of analysis. The daily mean wavefront speed had an IQR of 4.4–33.4 km/day. Wave 2 had a higher overall mean wavefront speed of 25.9 km/day (median = 20.6; IQR = 0–35.6 km/day). The daily mean wavefront had an IQR of 7.0–32.9 km/day. S1 Table provides daily mean and median wavefront speeds for each wave.

There was heterogeneity in the estimated daily wavefront speed across the edge of each wave. Fig 4 shows the speed of each wave at four dates leading up to each wave's infection peak. To understand changes in speed over the course of each wave, we calculated mean speed over three time periods: 63–43 days, 42–22 days, and 21–0 days prior to each wave's peak. These speed measures increased over the course of each wave, and for each time period, Wave 2 was faster than Wave 1, with mean speeds of 3.8 (median = 0; IQR = 0–0), 17.0 (median = 20.6; IQR = 0–20.6), and 33.9 km/day (median = 35.6; IQR = 20.6–54.4) and 6.8 (median = 0; IQR = 0–0), 19.8 (median = 20.6; IQR = 0-35.6), and 40.8 (median = 35.6; IQR = 20.6-54.4) km/day, respectively.

Because wavefronts varied in length, the percentage of the wavefront at maximal speed changed over time; to quantify this change, we calculated the length of the wavefront exceeding the median maximum speed over the study period. At its longest, the wavefront of Wave 1 had 7,017 kilometers moving at a speed greater than or equal to the overall median maximal speed at 18 days before its infections per 100,000 peak. In comparison, Wave 2 had a maximal wavefront of 9,157 kilometers at 9 days before its peak. The daily lengths of the wavefront moving faster than the median of the estimated daily maximum speeds are in S1 Table.

Animations of daily wavefront speeds are provided in S3 and S4 Movie **files,** for waves 1 and 2, respectively. All maps were generated using United States, state, and county borderlines maps in the public domain from the Census Bureau, which were downloaded through the R package Tigris [30]. The shapefile generated for this analysis with the population estimates and cumulative infection estimates can be found at: https://github.com/covidestim/waves/tree/waves-manuscript/Data/data-products.

## Discussion

In this study, we quantified the wavefront speed and the areal expansion rate of the first two major SARS-CoV-2 epidemic waves in the contiguous United States. By applying a Bayesian spatial modeling framework to spatially resolved infection estimates, we were able to provide new visualizations of wave movement and quantify and compare wave behavior. To produce these estimates, we adopted a definition of the wave front boundary corresponding to >190 infections per 100,000 per day. While this produces a wavefront that lags behind the first reported case at any given location, it provides a more stable measure of epidemic spread that corresponds to meaningful levels of community transmission.

Our findings suggest that the first and second waves differed in both their origin and expansion patterns. Wave 1 originated in parts of northeastern Montana, eastern North Dakota, and central South Dakota in September 2020 (**Fig 2B**-**2E** and S2 Movie). The estimated point of origin coincides with a motorcycle rally that brought over 460,000 attendees to Sturgis, South Dakota, in August 2020 and was later epidemiologically linked to elevated local COVID-19 rates and to the interstate spread of infection [33,34].

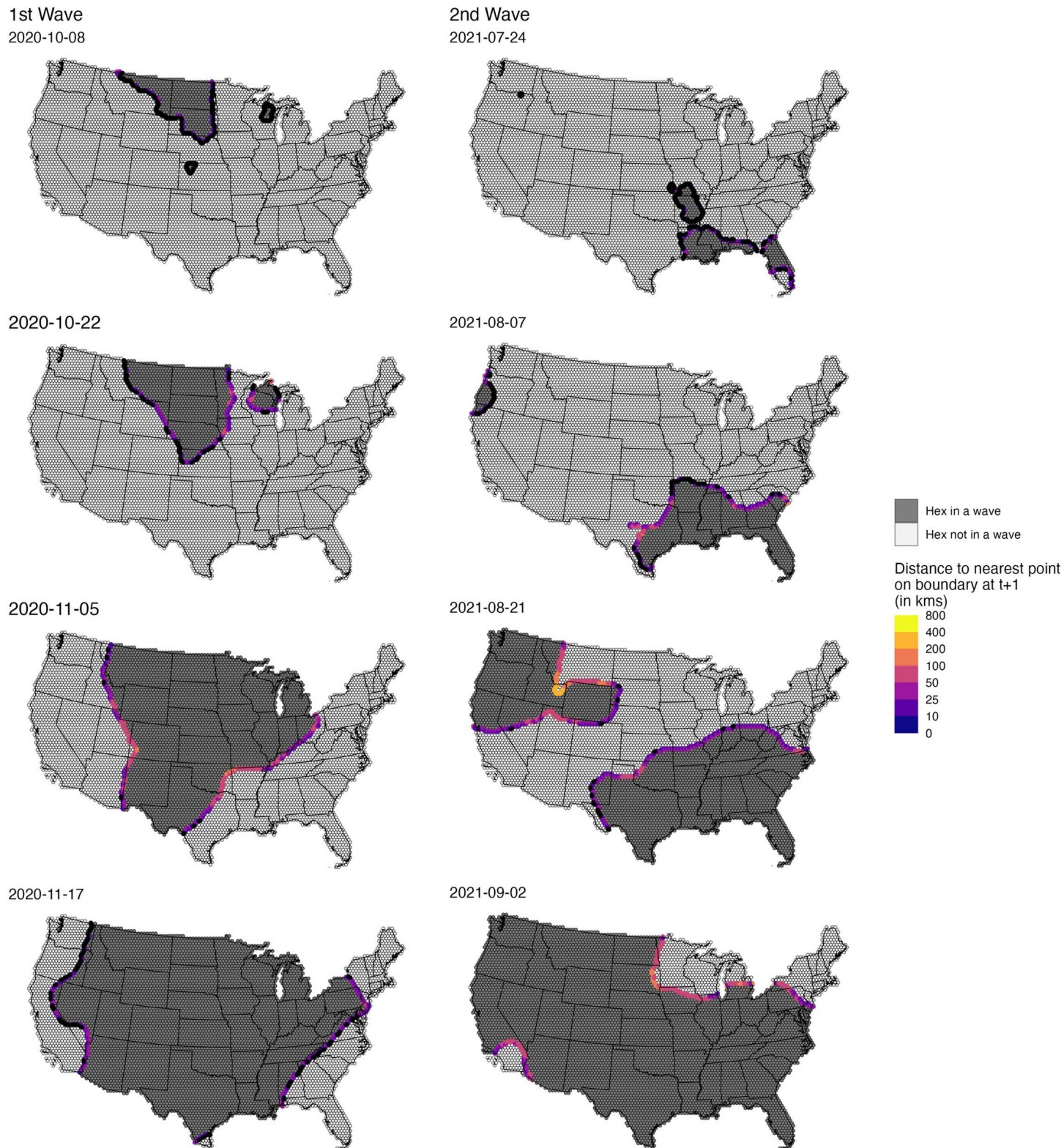

**Fig 4. Progression of the wave boundary measured as the distance to the nearest point on the boundary at t + 1 (in km/day).** Each panel shows a specific date; the ones in the right column are related to the first wave and the ones in the left column to the second wave. Vertices in black show

points with zero speed (i.e., their location is the same for t and t + 1. All maps were generated using United States, state, and county borderlines maps in the public domain from the Census Bureau, which were downloaded through the R package Tigris [30]. The shapefile generated for this analysis with the population estimates and cumulative infection estimates can be found at: https://github.com/covidestim/waves/tree/waves-manuscript/Data/data-prod-ucts. For a more detailed visualization of each wavefront, see the S3-S4 Movies.

In contrast, Wave 2 had multiple epicenters, beginning in the Ozarks (Fig 2F and S2 Movie) during a time when indoor music venues were reopening, and local masking policies were relaxed [35,36]. Shortly after, secondary high-incidence epicenters appeared in the Pacific Northwest. The decentralized and near-simultaneous emergence of multiple hotspots during Wave 2 aligns with earlier work demonstrating that both domestic and international travel networks have facilitated the rapid dispersal of viruses [6,37,38]. Wave 2 coincided with the emergence and national spread of the Delta variant (B.1.617.2), which became dominant in the United States during this period [18].

While we found that Wave 2 had both a higher maximal speed of wavefront movement and achieved its most significant areal expansion more quickly than Wave 1, the similarity in estimates of wavefront movement speed and the growth rate in wave size for these first two waves is striking. This similarity is remarkable considering the distinct virological, immuno-logical, and policy contexts in which each wave occurred. Wave 1 was due to wild type SARS-CoV-2 as well as the 'Alpha' variant [37,38]. Wave 2 was driven by the 'Delta' variant [17,18,35] and emerged at a time when approximately 50–60% of the US adult population had received at least one COVID-19 vaccination dose [39,40]. Regional and secular differences in non-pharmaceutical intervention policies, and differences in the levels and types of infection- and vaccine-induced popu-lation immunity at the time of these two waves [17], make the similarity between the movement patterns of these waves particularly surprising. In a previous analysis, we estimated state-level immunological protection against SARS-CoV-2 infection among US adults. At the onset of Wave 1 in September 2020, we estimated that ~5% of adults were protected against infection. This protection varied minimally between US regions, ranging from ~4% in the Midwest to ~7% in the Northeast. This immunological environment contrasted with that of Wave 2; in July 2021, an estimated ~34% of the US population was protected against SARS-CoV-2 infection, ranging from ~31% in the South to ~38% in the Northeast [41]. The consistent speed of expansion of the two waves, each in a different immunological context, may suggest that these expansion rates are relatively stable across immunological contexts. Alternatively, it could be that different factors (pop-ulation immunity, application of non-pharmaceutical interventions, pathogen characteristics, and underlying patterns of human mobility) had approximately offsetting effects in the two waves we examined [7,42].

Our effort to characterize and provide quantitative estimates of the speed of spread to epidemic waves required several simplifying assumptions. Importantly, we elected to use estimates of infections from a nowcasting model (rather than case notification data) as our input data. As diagnosis and case reporting varied markedly in quality and completeness over the epidemic [43], this choice allowed for more consistent input data. Another challenge we faced was systematic variability in county geography and population density across the contiguous United States. We also chose to distribute estimated infections on a hexagonal grid before estimating wavefront speed and areal patterns of wave expansion to address poten-tial bias associated with the irregular shapes and sizes of US counties (the smallest unit at which infection estimates were possible). Nevertheless, we could have introduced bias, since cases within counties were not randomly distributed; how-ever, this effect should be modest given the small sizes and large numbers of hexagons. Finally, we arbitrarily selected the threshold of infections per 100,000 to define wave membership but note that our sensitivity analyses showed that the estimated metrics describing wavefront speed and areal expansion rate were similar across alternative thresholds.

Factors that affect the diffusion and speed of epidemic waves are an area of substantial interest for other viral dis-eases [44], like influenza [45–47], Ebola (e.g., [48]), HIV [49], measles [50], and parasitic diseases such as malaria [51]. While our work on SARS-CoV-2 does not provide explanations for why each of these first two epidemic waves spread in the manner that they did once they were established, the development of methods to quantify the wavefront speed and rate of areal expansion of epidemic waves, such as those we employed here, is a necessary first step. Related work

has evaluated the quantification of wave dynamics for point pattern disease data [52–54]and from trend surface analysis [55–57]. Further research to explore how the pathogen, environment, and host characteristics affect the speed and pattern of epidemic expansion may improve epidemic predictions and assist in the planning and deployment of spatially defined interventions.

## Supporting information

**S1 Appendix. Supplementary Material.**
(DOCX)

**S1 Fig. Empirical cumulative density function (ECDF) of the risk surface values.** The ECDF shows that a threshold of 190 infections per 100,000 is indicated by the vertical dashed line at which the ECDF crosses the 75th percentile. The ECDF for the lower and upper bounds is shown in grey.
(TIF)

**S2 Fig. Sensitivity analysis of areal wave expansion to the threshold values.** Areal wave expansion (km2/day) for different thresholds of infection per 100,00 in the progression calculation of the surfaces. Panel **A** is built with a threshold of 127 or more infections per 100,000; panel **B** is built with a threshold of 233 infections per 100,000. As in **Fig 3C**, we observe a maximal speed and a steep decrease after the peak, and the second wave had a higher invasion speed and encompassed a larger area at peak than the first wave.
(TIF)

**S3 Fig. Infection per 100,000 surface on a continuous scale of values.** As expected, the wave-like pattern holds independently of the scale to be displayed, and as being an output of spatial smooth model, the continuous scale gives a less defined border to the risk surface expansion. All maps were generated using United States, state, and county borderlines maps in public domain from the Census Bureau which were downloaded through the R package Tigris [30]. The shapefile generated for this analysis with the population estimates and cumulative infections estimates can be found at: https://github.com/covidestim/waves/tree/waves-manuscript/Data/data-products.
(TIF)

**S4 Fig. Infection per 100,000 with a threshold equal to the mean of the risk values distribution (127 infections per 100,000).** With a lower threshold showing on the map, the spread process seems to happen faster. All maps were generated using United States, state, and county borderlines maps in public domain from the Census Bureau which were downloaded through the R package Tigris [30]. The shapefile generated for this analysis with the population estimates and cumulative infections estimates can be found at: https://github.com/covidestim/waves/tree/waves-manuscript/Data/data-products.
(TIF)

**S1 Table. Daily estimates of wavelength, areal growth, and speed for each wave.**
(DOCX)

**S1 Movie. Wave 1 spreading sequential maps.**
(MOV)

**S2 Movie. Wave 2 spreading sequential maps.**
(MOV)

**S3 Movie. Wave 1 wavefront speed sequential maps.**
(MOV)

**S4 Movie. Wave 2 wavefront speed sequential maps.**
(MOV)

## Author contributions

**Conceptualization:** Rafael Lopes, Melanie H. Chitwood, Nicolas A. Menzies, Ted Cohen, Nicole A. Swartwood.

**Data curation:** Fayette Klaassen.

**Funding acquisition:** Joshua A. Salomon, Nathan D. Grubaugh, Ted Cohen.

**Investigation:** Rafael Lopes, Nicole A. Swartwood.

**Methodology:** Rafael Lopes, Nicolas A. Menzies, Joshua L. Warren, Ted Cohen, Nicole A. Swartwood.

**Supervision:** Joshua A. Salomon, Nicolas A. Menzies, Joshua L. Warren, Nathan D. Grubaugh, Ted Cohen, Nicole A. Swartwood.

**Visualization:** Rafael Lopes, Yu Lan, Nicolas A. Menzies, Nicole A. Swartwood.

**Writing – original draft:** Rafael Lopes, Nicolas A. Menzies, Nicole A. Swartwood.

**Writing – review & editing:** Rafael Lopes, Yu Lan, Melanie H. Chitwood, Fayette Klaassen, Joshua A. Salomon, Nicolas A. Menzies, Joshua L. Warren, Nathan D. Grubaugh, Ted Cohen, Nicole A. Swartwood.

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
