## [Decision Letter · Decision Letter 0]

16 Mar 2025

Quantifying the spatiotemporal dynamics of the first two epidemic waves of SARS-CoV-2 infections in the United States

PLOS Computational Biology

Dear Dr. Lopes,

Thank you for submitting your manuscript to PLOS Computational Biology. After careful consideration, we feel that it has merit but does not fully meet PLOS Computational Biology's publication criteria as it currently stands. Therefore, we invite you to submit a revised version of the manuscript that addresses the points raised during the review process.

Please submit your revised manuscript within 60 days May 16 2025 11:59PM. If you will need more time than this to complete your revisions, please reply to this message or contact the journal office at ploscompbiol@plos.org. Please include the following items when submitting your revised manuscript:

We look forward to receiving your revised manuscript.

Kind regards,

Claudio José Struchiner, M.D., Sc.D.

Academic Editor

PLOS Computational Biology

Benjamin Althouse

Section Editor

PLOS Computational Biology

**Additional Editor Comments:**

The reviewers have identified important methodological concerns and emphasized the necessity of elucidating the practical implications of the study's findings. Notably, the definitions of "wave" and "speed" lack precision and clarity, potentially limiting their utility. Addressing these issues would significantly enhance the manuscript's scientific rigor.

**Journal Requirements:**

https://journals.plos.org/ploscompbiol/s/figures. Please also ensure that figure 3 is labeled.

3) We have noticed that you have uploaded Supporting Information files, but you have not included a complete list of legends. Please add a full list of legends for your Supporting Information files (Supplementary Movies) after the references list.

Potential Copyright Issues:

i) Figures 1, 2, S3, S4, and S5. Please (a) provide a direct link to the base layer of the map (i.e., the country or region border shape) and ensure this is also included in the figure legend; and (b) provide a link to the terms of use / license information for the base layer image or shapefile. We cannot publish proprietary or copyrighted maps (e.g. Google Maps, Mapquest) and the terms of use for your map base layer must be compatible with our CC BY 4.0 license.

5) We note that your Data Availability Statement is currently as follows: "All relevant data are within the manuscript and its Supporting Information files." Please confirm at this time whether or not your submission contains all raw data required to replicate the results of your study. Authors must share the “minimal data set” for their submission. PLOS defines the minimal data set to consist of the data required to replicate all study findings reported in the article, as well as related metadata and methods (https://journals.plos.org/plosone/s/data-availability#loc-minimal-data-set-definition).

7) Please ensure that the funders and grant numbers match between the Financial Disclosure field and the Funding Information tab in your submission form. Note that the funders must be provided in the same order in both places as well." Currently, "Council of State and Territorial Epidemiologists (CSTE)" is missing from the Funding Information tab.

**Reviewers' comments:**

Reviewer's Responses to Questions

Reviewer #1: The authors present a very interesting work on the speed of SARS-CoV-2 spread across the United States. The work is of interest to the wider public as (i) it can be used to inform public policy both in terms of surveillance and response and (ii) it can easily be applied to other viruses as such methodologies are highly scarce. However, a few things need more clarity and attention, as the manuscript in very succinct at times.

1- It would be good to include some information in the introduction about the different waves in the US: epidemiological scenario, variants, interventions, etc. This is really important to understand the results and it is only really brought up in the discussion.

2- It is not clear to me how the waves were selected. I do understand that a threshold for the number of infections was used for that. However, wave 1 is only considered to have started in September 2020, when the wildtype was already circulating in the united states for at least 7 months. As a reader, it would have been more interesting to see a characterisation of the epidemic in the country since its introduction and first original wave back in March. This would reveal an interesting perspective on the role of different areas of the country in spreading the virus in a completely naive population, which is not true for the two waves analysed here. It is also confusing that the large wave in September is called wave 1 when from the plots there were two prior waves, much smaller, but still very epidemiologically important at the time. For instance, a close look at the very first wave March to June could have revealed the potential impact of interventions in decreasing the spread through a potential reduction in the speed.

3- A suggestion could also be to define waves based on variant surveillance. For instance, it is assumed that both waves started from one focus of spread. However, while the second wave probably overlaps with the introduction of delta, wave 1 had already seen 7 months of wildtype spread. A model using genomic data and case counts could help pinpoint the start of a new way. (not mandatory)

4- While from the maps it is very clear that wave 1 did start in the midwest and then radially spread across the country, I believe that wave 2 is better described as a wave caused by several origins, which could well describe a wave caused by the introduction of a new variant in multiple areas or with multiple variants circulating in different areas of the country. As noted in figures 2F-H there seems to have been multiple origins for wave two: the Ozarks, Wyoming, Nevada and Oregon.

5- Speed or velocity of spread are normally represented in km/day as they describe the movement in linear direction rather than area, you would probably be more appropriate to either represent speed as km/day, or call the measure being report as a diffusion coefficient or rate, km2/day.

6- 580,000 km2/day would be around 760 km/day. This would be equivalent to spread form New York to Columbus in one single day. Although that is probably correct due to the high aerial connectivity in the USA, I think it would be probably more interesting to understand what the median and mean speed of spread/diffusion coefficients are as spread can be highly skewed and long-tailed, with long distance movements having some impact but normally less frequency than short distance movements. What is the distribution of the estimates for the wavefront velocity increase? Finally, how did the authors handle outliers, i.e., days in which the spread was extremely high or extremely low?

7- Although the speed of spread is the main finding in the manuscript, there is little description on how the calculation was actually performed. For instance, how was distance measured here? What kind of distance is being used, geodesic? Which point in the grid was used, centroid?

8- Were there any regional differences in the speed of spread? Maps coloured by the speed of spread/diffusion rate would give us a greta information on local variation in virus spread.

9- Finally, while the text refers to several panels within figure 3, there is only one graph being presented in the actual figure. Probably the other figures would be needed to understand the full context of the manuscript.

Reviewer #2: In their article, López et al. present an innovative approach to mapping the spatial spread of SARS-CoV-2 infections across the United States, using the BYM2 model to generate daily infection per capita surfaces, which were later used to calculate the speed of wave expansion. Overall, I found the article well-written and easy to follow. The methodology is clearly described, and the visualizations—both the plots and movies—are nicely done and effective in illustrating their results.

However, I have a few minor comments:

1. The authors used an area-weighted approach to distribute population estimates from CBGs to hexagons, assuming a constant population throughout the analysis period. However, could they have instead used a population raster, with high-resolution gridded population data, to assign a more precise and spatially realistic population estimate per hexagon? Same thing with the estimates of infections. Why did you choose an area-weighted approach?

2. In lines 178-186 the authors note that while Wave2 showed a higher, but overall similar, maximal speed of expansion than Wave1, they argue that human mobility played a more significant role than viral transmissibility, immunity, and policy changes. However, given that (i) multiple studies have demonstrated that the Delta variant is more transmissible than both the wild-type and Alpha variants, (ii) international and domestic travel restrictions were more relaxed in Wave 2 compared to Wave 1, (iii) NPIs were also more relaxed during the Wave 2, shouldn’t the Wave2 have spread more rapidly then as written in lines 33-34? Maybe the lack of substantial difference in the speed of expansion could be attributed to the (likely) increased vaccine coverage and naturally acquired immunity in Wave2? Could the authors clarify the reasoning behind their conclusion that human mobility had a stronger influence than these other variables?

3. The GitHub repository does not appear to be accessible, please make it available before publication.

Reviewer #3: Title: Quantifying the Spatiotemporal Dynamics of the First Two Epidemic Waves of SARS-CoV-2 Infections in the United States

Manuscript Number: PCOMPBIOL-D-25-00080

# Overall Assessment

This manuscript presents a quantitative analysis of the spatiotemporal dynamics of the first two major COVID-19 infection waves in the United States. The study employs a BYM2 spatial model to estimate an infection surface and uses it to evaluate the speed and spatial extent of viral spread. The manuscript is well-structured, and the methods are sound. However, the analysis is broad and lacks deeper insights. Some areas require further clarification and improvement, as outlined below.

# Key Findings

The research provides effective visualizations of the spread of SARS-CoV-2. According to the Author Summary, this is the main goal of the study. The authors also produce global estimates for disease wave spread speed during two major waves in the U.S.

# Strengths

- The visualizations are clear and well-executed.

# Weaknesses

- The definitions of wave and speed are vague.

- The concept of speed, as derived, has limited applicability.

# Specific Comments

- I could not access the GitHub repository: [https://github.com/covidestim/waves](https://github.com/covidestim/waves).

Line 33-34: "driven by multiple, non-contagious sources of infection" – A source is needed to support this statement.

Line 54: "Efforts to quantify the speed and spatial extent of waves have been limited" – While this is true, there have been attempts to quantify disease spread. Some of these studies focus on point pattern analysis of disease spread, which is worth mentioning. Some are mentioned at the end of this review.

Line 80: "each enclosing 64.75 square kilometers" – The reason for this choice should be explained. Is it based on a minimum area threshold for U.S. census block groups (CBGs)? Are the estimates sensitive to this choice?

Line 81: "We employed an area-weighted approach" – Area weighting might introduce bias since areas with little or no population may receive a disproportionate share of the estimate. Consider referencing ESRI's methodology for population estimation based on land-use data, which classifies land from densely populated to rural and distributes census estimates accordingly. This would likely produce more reliable population estimates.

Line 97-98: "for more details on the priors used, see the Supplementary Material" – The supplementary material only states that PC-priors were used. If the hyperparameter selection followed INLA’s default choice, this should be explicitly mentioned.

Line 107: The definition of wave is vague. The study defines waves spatially, yet Wave 1 and Wave 2 are described in temporal terms. The definition should encompass both spatial and temporal characteristics.

Line 118: The definition of speed is simplistic and may not be informative. Specifically:

- The definition of speed is global, which presents issues. As the visualizations and author comments indicate, regional heterogeneity is of great interest. Global speed estimates do not provide insight into local events.

- Events such as the motorcycle rally (Line 165) and the relaxation of mask policies (Line 169) should be examined in the context of local disease spread estimates.

- Notably, the manuscript does not use speed estimates to derive meaningful conclusions. Speeds are merely computed and compared to the national curve peak and to one another.

Line 163: Regarding the estimated origins of the waves, it is important to clarify that these estimations were based on visual inspection of the graphics. No explicit coordinate-based epicenter estimation was performed.

Figure 1-D: Some regions have no recorded infections, but the smoothed surface produced by the BYM2 model extends disease spread into these areas, which affects the speed calculation. In a discrete spatial model, this issue can be addressed by removing these regions and updating the adjacency matrix accordingly. How would this adjustment impact the speed estimates? One possible approach to selecting regions for removal is identifying areas with dense forest coverage or sparsely populated rural zones (as mentioned in comments above).

Figure S5: The figure is confusing. The description refers to "dates for the speed of expansion," but no speed information is presented. A more accurate description would be "wave boundary limits at various dates up to the national curve peak date."

# Additional Methodological References

In point pattern models using partial differential equations (PDEs), the definition of velocity is straightforward. Some relevant works on estimating disease spread through point patterns include:

- Velocities for spatio-temporal point patterns ([https://doi.org/10.1016/j.spasta.2018.12.007](https://doi.org/10.1016/j.spasta.2018.12.007))

- Estimating velocities of infectious disease spread through spatio-temporal log-Gaussian Cox point processes ([https://arxiv.org/abs/2409.05036](https://arxiv.org/abs/2409.05036))

A related approach to the authors’ method is trend-surface analysis (TSA), where a surface is fitted to reported cases and speeds are estimated from the surface. Relevant studies include:

- Estimating front-wave velocity of infectious diseases: a simple, efficient method applied to bluetongue ([https://doi.org/10.1186/1297-9716-42-60](https://doi.org/10.1186/1297-9716-42-60))

- Spatial diffusion of raccoon rabies in Pennsylvania, USA ([https://doi.org/10.1016/s0167-5877(99)00005-7](https://doi.org/10.1016/s0167-5877(99)00005-7))

Other relevant works can be found within the references.

**Have the authors made all data and (if applicable) computational code underlying the findings in their manuscript fully available?**

Reviewer #1: Yes

Reviewer #2: **No:** The GitHub repository does not appear to be accessible: https://github.com/covidestim/waves

Reviewer #3: **No:** The GitHub repository could not be found.

PLOS authors have the option to publish the peer review history of their article (what does this mean? ). If published, this will include your full peer review and any attached files.

**Do you want your identity to be public for this peer review?** For information about this choice, including consent withdrawal, please see our Privacy Policy .

Reviewer #1: No

Reviewer #2: No

Reviewer #3: No

**Figure resubmission:**
---

## [Decision Letter · Decision Letter 1]

19 Aug 2025

PCOMPBIOL-D-25-00080R1

Quantifying the spatiotemporal dynamics of the first two epidemic waves of SARS-CoV-2 infections in the United States

PLOS Computational Biology

Dear Dr. Lopes,

Thank you for submitting your manuscript to PLOS Computational Biology. After careful consideration, we feel that it has merit but does not fully meet PLOS Computational Biology's publication criteria as it currently stands. Therefore, we invite you to submit a revised version of the manuscript that addresses the points raised during the review process.

Please submit your revised manuscript within 30 days Oct 19 2025 11:59PM. If you will need more time than this to complete your revisions, please reply to this message or contact the journal office at ploscompbiol@plos.org. Please include the following items when submitting your revised manuscript:

We look forward to receiving your revised manuscript.

Kind regards,

Claudio José Struchiner, M.D., Sc.D.

Academic Editor

PLOS Computational Biology

Benjamin Althouse

Section Editor

PLOS Computational Biology

**Journal Requirements:**

1) We have noticed that you have uploaded Supporting Information files, but you have not included a complete list of legends. Please add a full list of legends for your Supporting Information files (Supplementary movies) after the references list.

2) Your current Financial Disclosure indicated receiving funds; however, you stated that "The authors received no specific funding for this work." Please amend your Financial Disclosure statement and indicate the full and correct funding information for your study.

3) Please ensure that the funders and grant numbers match between the Financial Disclosure field and the Funding Information tab in your submission form. Note that the funders must be provided in the same order in both places as well. Currently, "Cooperative Agreement NU38OT000297" and " the Council of State and Territorial Epidemiologists (CSTE)" are missing from the Funding Information tab.

**Reviewers' comments:**

Reviewer's Responses to Questions

Reviewer #1: Thank you very much for addressing the points raised in the previous round of reviews. The manuscript has clearly improved and is now much clearer. However, I still have a few concerns that I believe warrant further attention:

1-Presenting both the area and speed of expansion is a good idea. The estimated area of expansion—around 1600 km²/day or roughly 40x40 km/day—appears plausible and consistent with the figures. However, the reported speed of spread, especially for wave 2, seems too high. If speed is calculated as the straight-line distance between the wavefronts of consecutive days, then a median area of 1600 km² and a median distance of 748 km/day (wave 2) would imply a corresponding orthogonal component of only ~2 km, which seems inconsistent. Furthermore, according to the videos, the virus took around 45 days to spread across the country, whereas the median speed suggests it would have gone from New York to Seattle in under a week. Most grid lines until second 00:19 in the video are black or purple (indicating speeds ≤200 km/day), with high speeds only appearing in the final seconds. This raises doubts about the plausibility of a median speed of ~700 km/day.

2-Relatedly, the speed values reported in Supplementary Table 5 for wave 2 do not appear to match the wave 2 video. Most estimates in the table are above 500 km/day, with many above 900 km/day (orange in the video legend), whereas the video mostly shows black and purple lines, again suggesting speeds ≤200 km/day. This discrepancy should be addressed.

3-Are the colours in Figure 3 correct? They seem to be inverted compared to the supplementary figures. If they are correct, then for most of the timeline, wavefront 1 spread faster than wavefront 2, with wavefront 2 only briefly overtaking at the peak. This is the opposite of what is seen for the other thresholds. If the colours are correct, this should be clarified in the limitations section, highlighting the sensitivity of the results to the choice of threshold.

4-From a policy perspective, it would be helpful to clarify that the analysis reflects when districts reached the defined threshold—not necessarily when the first case was detected. While this is clear from a methodological standpoint, emphasizing it in more accessible terms would aid broader interpretation.

5-The introduction could still benefit from more context about the two waves, including non-pharmaceutical interventions and vaccine rollout status. These contextual details are critical for interpreting the findings, especially for readers less familiar with the field or in future contexts.

6-The manuscript discusses geographic variation but omits the temporal aspect of spread. Highlighting that the initial spread was geographically limited but temporally faster closer to the peak would strengthen the interpretation of the dynamics.

7- Figure labels should be revised, as areal wavefront is sometimes mislabelled as speed. Resolution of figures needs to be improved.

Reviewer #2: I am satisfied with the author’s revisions and find that the updated document address prior concerns.

Reviewer #3: Thank you for the thorough revision of my comments. The new version addresses my substantive requests. I have some minor editorial fixes to finalize the manuscript

Line 102: add space in “μis” → “μ is.”

Line 103: add space in “ϵ( )are” → “ϵ( ) are.”

Line 184 (Results, “Wave speed”): fix “overal” → “overall.”

Line 234 (Discussion): resolve the fragment “adds explanatory factors to the stability observed in diffusion and speed of spread of each wave.” Suggested replacement:

“This adds explanatory factors for the stability observed in diffusion and in the speed of spread of each wave.”

With these small edits, I am satisfied with the revision.

**Have the authors made all data and (if applicable) computational code underlying the findings in their manuscript fully available?**

Reviewer #1: Yes

Reviewer #2: Yes

Reviewer #3: Yes

PLOS authors have the option to publish the peer review history of their article (what does this mean? ). If published, this will include your full peer review and any attached files.

**Do you want your identity to be public for this peer review?** For information about this choice, including consent withdrawal, please see our Privacy Policy .

Reviewer #1: No

Reviewer #2: No

Reviewer #3: No

**Figure resubmission:**
---

## [Decision Letter · Decision Letter 2]

16 Jan 2026

PCOMPBIOL-D-25-00080R2

Quantifying the spatiotemporal dynamics of the first two epidemic waves of SARS-CoV-2 infections in the United States

PLOS Computational Biology

Dear Dr. Lopes,

Thank you for submitting your manuscript to PLOS Computational Biology. After careful consideration, we feel that it has merit but does not fully meet PLOS Computational Biology's publication criteria as it currently stands. Therefore, we invite you to submit a revised version of the manuscript that addresses the points raised during the review process.

We look forward to receiving your revised manuscript.

Kind regards,

Claudio José Struchiner, M.D., Sc.D.

Academic Editor

PLOS Computational Biology

Benjamin Althouse

Section Editor

PLOS Computational Biology

**Reviewers' comments:**

Reviewer's Responses to Questions

**Comments to the Authors:**

Reviewer #1: Thank you to the authors for addressing the previous comments so thoroughly. I am very pleased with the revisions and the clear improvements in the manuscript. The updated estimates are now much more interpretable and can be confidently used to draw conclusions about virus spread, its drivers, and consequences. I have only three minor final comments.

1. I understand the rationale for focusing on the mean given that the median is often zero. However, the median still provides valuable information about the underlying distribution that is not captured by the mean alone. I therefore suggest reporting the median together with the interquartile range (IQR), alongside the mean, at least in the text, to give a more complete picture of the estimates.

2. For Figure 3, please clarify on the y-axis whether the values shown correspond to the mean or the median. In addition, I recommend expanding the legend to more explicitly specify the infection type, time period, and a brief description of the general estimation approach used.

3. While I understand that providing detailed explanations for differences between the two waves is not the primary aim of the study, this discussion would, in my view, substantially enhance the manuscript’s impact. In particular, it would be helpful to include in the discussion some quantitative context regarding population immunity at the time of Wave 2. Based on publicly available data (e.g. estimates suggesting ~50% coverage with two vaccine doses at that time), immunity levels might not have been sufficient to substantially reduce wavefront speed—especially given the immune escape properties of the Delta variant (https://usafacts.org/visualizations/covid-vaccine-tracker-states/). I agree with the authors that human mobility likely played the dominant role, which could explain the similar mean speeds of spread across waves (and potentially even more similar medians). However, as currently written, the discussion could be interpreted as suggesting that vaccination and non-pharmaceutical interventions (NPIs) had little to no effect. It is important to emphasize that vaccination uptake and adherence to NPIs were highly heterogeneous across the US. Therefore, the apparent lack of impact on spread velocity may reflect these inequalities in uptake and compliance, rather than the intrinsic effectiveness of these measures. This distinction is important, particularly given strong evidence from other studies that vaccination and NPIs are highly effective when widely and consistently implemented. I appreciate that this was likely not the authors’ intention, but a clarification here would strengthen the interpretation.

4. Please check references. Is reference 40 correct? I was expecting a publication about immunity in the US but it is instead about B.1.1.7 in the UK.

**Have the authors made all data and (if applicable) computational code underlying the findings in their manuscript fully available?**

Reviewer #1: None

PLOS authors have the option to publish the peer review history of their article (what does this mean? ). If published, this will include your full peer review and any attached files.

**Do you want your identity to be public for this peer review?** For information about this choice, including consent withdrawal, please see our Privacy Policy .

Reviewer #1: No

**Figure resubmission:**
---

## [Editor Report · Decision Letter 3]

4 Feb 2026

Dear Lopes,

We are pleased to inform you that your manuscript 'Quantifying the spatiotemporal dynamics of the first two epidemic waves of SARS-CoV-2 infections in the United States' has been provisionally accepted for publication in PLOS Computational Biology.

Best regards,

Claudio José Struchiner, M.D., Sc.D.

Academic Editor

PLOS Computational Biology

Benjamin Althouse

Section Editor

PLOS Computational Biology

---

## [Editor Report · Acceptance letter]

PCOMPBIOL-D-25-00080R3

Quantifying the spatiotemporal dynamics of the first two epidemic waves of SARS-CoV-2 infections in the United States

Dear Dr Lopes,

I am pleased to inform you that your manuscript has been formally accepted for publication in PLOS Computational Biology. Your manuscript is now with our production department and you will be notified of the publication date in due course.

With kind regards,

Livia Horvath
